# Portable digital devices for paediatric height and length measurement: A scoping review and target product profile matching analysis

Tasmyn Soller[1]*, Shan Huang[1], Sayaka Horiuchi[1], Alyce N. Wilson[1], Joshua P. Vogel[1,2]

**1** Maternal, Child and Adolescent Health Program, Burnet Institute, Melbourne, Victoria, Australia, **2** School of Public Health and Preventive Medicine, Monash University, Melbourne, Victoria, Australia

* tasmynsoller@gmail.com

## Abstract

### Background

Routine anthropometry of children, including length/height measurement, is an essential component of paediatric clinical assessments. UNICEF has called for the accelerated development of novel, digital height/length measurement devices to improve child nutrition and growth surveillance programs. This scoping review aimed to identify all digital, portable height/length measurement devices in the literature or otherwise available internationally. We also assessed identified devices against the UNICEF Target Product Profile (TPP) to identify those of highest potential for clinical and public health use.

### Method

We searched four databases (Medline, Embase, CINAHL and Global Health) and the grey literature between 1st January 1992 and 2nd February 2023. We looked for studies or reports on portable, digital devices for height or length measurement in children up to 18 years old. Citations were screened independently by two reviewers, with data extraction and quality assessment performed in duplicate and disagreements resolved. Devices were evaluated and scored against the 34 criteria of the UNICEF TPP.

### Results

Twenty studies describing twelve height/length measurement devices were identified, most of which used prospective validation designs. Additional devices were found in the grey literature, but these did not report key performance data so were not included. Across the twelve devices, only 10 of 34 UNICEF criteria on average could be fully assessed. Six met UNICEF's ideal accuracy standard and one device met the minimum accuracy standard. The Leica DistoD2 device scored highest (41%), followed by Autoanthro in a controlled environment (33%) and GLM30 (32%). These devices may be high potential for further assessment and development, though further research is required.

**Data Availability Statement:** All relevant data are within the manuscript and its Supporting Information files.

**Funding:** The authors received no specific funding for this work.

**Competing interests:** The authors have declared that no competing interests exist.

## Conclusion

While 12 portable, digital devices exist for child height/length measurement, insufficient data are available to fully assess whether they meet the industry's needs. Although some devices show promise, further research is needed to test the validity of these devices in varying contexts, and continued development and commercialization will be important to improve reliability and precision of these devices for widespread use.

## Introduction

Routine anthropometry of infants and children, including weight, length/height and head circumference are essential components of nutritional assessments [1]. Measurement of height (standing up) for children aged 2 years and older and length (lying down) for children aged less than 2 years allows for detection and monitoring of stunting, wasting and obesity in children globally [2, 3]. Stunting and wasting are both forms of under-nutrition, and are significant impediments to human social and economic development. Stunting, defined as more than two standard deviations below the median height-for-age/length-for-age, affects approximately 162 million children under 5 years worldwide [4]. Stunting has long-term physical, cognitive, educational and economic consequences for individual children, as well as their families and communities [5]. Wasting is defined as more than two standard deviations below the median weight-for-height/weight-for-length and indicates recent and severe weight loss [4]. If undiagnosed or untreated, wasting is associated with a higher risk of death [4]. Conversely obesity—a body mass index greater than 3 standard deviations above the WHO growth standard median—also has significant health implications for children. Globally, 39 million children under 5 and 340 million children over 5 are overweight or obese [6]. Obesity is itself a non-communicable disease, and a risk factor for the development of other chronic conditions which lead to premature death and disability [6].

Given the physical, cognitive and developmental implications of poor nutrition, accurate anthropometric assessment of children is an essential public health practice, and ensures stunting, wasting and obesity are detected and treated early [1]. In addition, routine anthropometric measurements form the basis of critical health indicators for monitoring children's health at a population level, which can assist in evaluating whether health and nutritional interventions are working, and also to assess global progress towards the Sustainable Development Goals [2, 3].

Measurement boards (stadiometers) are currently the gold-standard tool for obtaining child height and length, including in community-based child health surveillance activities [7]. While they have been in widespread use for decades, they are cumbersome and heavy [8]. The process of manual data collection also increases the risk of transcription errors, negatively affecting data accuracy. A digital, accurate and precise measurement device would thus provide significant advantages for clinical assessment and nutritional surveillance of children. If such a device were portable, lightweight, and reliable, with simple data storage and transfer abilities, it would make child health assessments easier. This is especially pertinent in resource-limited settings, where the large, cumbersome measurement board can be particularly difficult to transport and use, and reliable data transcription and record-keeping can be challenging.

UNICEF has highlighted the need for innovation in developing height/length measurement devices that can improve nutritional surveillance and clinical assessment of children. In 2017, UNICEF published a new Target Product Profile (TPP) for digital devices for measuring child height/length, which describes the required operational, performance, usability and

commercialization requirements [9]. This TPP was intended to inform and guide new product development by commercial entities, yet it also provides an objective set of criteria by which a novel device could be evaluated. That is, assessment of device capability against TPP criteria can help identify those devices that best meet pre-specified clinical and operational needs, or the most promising candidates for further development. A TPP-based matching approach has been used previously to identify high-potential candidates for new obstetric medicines [10].

While a number of digital devices have been tested or are commercially available, no previous review has assessed the number, characteristics and quality of these devices. Furthermore, no review has evaluated whether tested devices meet clinical and public health requirements, which is essential to consider in order to bridge the gap between device development research and real-world needs. We aimed to identify and evaluate all portable digital devices for child height/length measurement. We also aimed to assess all devices against the criteria within the UNICEF TPP, in order to identify high-potential devices for further development or implementation.

## Methods

### Study design

This was a scoping review conducted in accordance with the Joanna Briggs Institute Methodology for Scoping Reviews and the Preferred Reporting Items for Systematic Reviews and Meta-Analyses extension for Scoping Reviews (PRISMA-ScR) standards, using the Arksey and O'Malley methodological framework [11, 12]. The research question and study selection criteria and analytic framework were developed using multidisciplinary expertise and consultation within the scoping team to guide the search criteria and applicability of data for extraction. An analytic framework was developed using the 34 criteria of the UNICEF TPP (Fig 1), against which all identified devices that met the eligibility criteria were assessed [9]. The review protocol was registered online through the Open Science Foundation (OSF) [13]. As a scoping review of publicly available data, ethics approval was not required. No patients or members of the public were involved in the design or conduct of this review.

### Eligibility criteria

Eligible studies were primary research studies that used any study design and conducted in any country or setting, provided that the study involved the use of a digital height/length measurement device in children under 18 years. Relevant product-related information, device manuals and webpages for any identified devices were also recovered during the grey literature search. Devices found in the grey literature only were included in the analysis if there was evidence of assessment of the device's reliability and precision, which are essential criteria as per UNICEF's TPP. We searched the literature from 1 January 1992 onwards, which captured 30 years of data, allowing for a thorough evaluation of contemporary products. Studies on conventional measurement tools only (i.e., height board/stadiometers) were not included. Systematic, scoping or narrative reviews were not eligible, but if they pertained to the review topic of interest, the references were checked for any potentially eligible studies.

### Literature searching and eligibility assessment

We searched four databases—Medline, Embase, CINAHL, and Global Health–on 18[th] January 2022 and updated the search on 2[nd] February 2023 (Appendix 1 in S1 Appendix). We decided on these databases following exploratory searches and consultation with an information specialist. This specialist also developed search strategies for each database, combining relevant

| | Ideal requirement | Explanation |
|---|---|---|
| **A) Operational/functional requirements** | | |
| 1 | Provides reliable length/height measurements of human subjects | Measurements have accuracy and precision within 0.2cm (the current standard of UNICEF endorsed devices) |
| 2 | Used for growth monitoring | Must measure length/height in children |
| 3 | Can be stored and used in a wide range of conditions | Must be able to be stored in hot, cold, dry, wet and dusty conditions. It can be moved in and out of vehicles, carried over distances on harsh and bumpy terrain. |
| 4 | Can measure height (standing) and length (lying) of subjects | - |
| 5 | Rechargeable battery | Battery must be rechargeable and last a minimum of 48 hours |
| 6 | Auto shut-off capabilities | If not in use it will auto-shut off to save power |
| 7 | Notification at low power | At low power remaining time of functionality indicated |
| 8 | Automatic calibration | Does not need calibration or this is done automatically |
| 9 | Digital display | Digital display in cm with one decimal point |
| 10 | Good storage capacity and connects via Wi-Fi and/or Bluetooth | Storage capacity of 5GB and supports windows compatible data download |
| 11 | Quick assembly | Takes less than one minute to assemble |
| **B) Performance requirements:** | | |
| 1 | Accurately measures static object | Within 1mm |
| 2 | Accurately measures human subjects | Within 1mm |
| 3 | Precise | Within 1mm |
| 4 | Good range | At least 30-215cm |
| 5 | Immediate results | <1 second |
| **C) Product requirements:** | | |
| 1 | Long storage life | Minimum 24 months |
| 2 | Long operational life | Minimum 5 years |
| 3 | Operates across wide ranging climactic conditions | Operates between -10 and -45 degrees Celsius and up to 80% humidity |
| 4 | Durable | It must be able to withstand the devices intended use without comprising functionality or accuracy, meeting standards such as EN 62262 IK 09 or IEC 60529 |
| 5 | Child friendly | It must be designed to avoid distress or harm by using factors such as soft edges, colours and illustrations |
| 6 | Minimal maintenance | No maintenance required with exception of light cleaning |

| | Ideal requirement | Explanation |
|---|---|---|
| **D) User requirements:** | | |
| 1 | Portable | The device is easily and comfortably transported from one site to another by the use of handles or backpacks. Maximum 2kg. |
| 2 | Assembly can be managed by one person | Assembly can be managed by one person without tools |
| 3 | Can be operated by only one person | Subject can be positioned, measured and readings documented accurately by one person |
| 4 | Clear output display | Backlit LCD screen with anti-glare screen and high contrast |
| 5 | Automatic data download | Data is automatically transferred to preferred device, even with low power. Connectivity range within 10m. |
| 6 | Simple training only and can be operated by non-technical users | Less than one day of training needed to be understood by non-technical users |
| **E) Supply chain requirements:** | | |
| 1 | Easily packable | High packing density and compatible with standard palate sizes. |
| 2 | Low environmental footprint | Device including packaging must be made of environmentally friendly and safe materials, preferably recyclable. |
| **F) Commercialization requirements:** | | |
| 1 | Meets international regulatory requirements and standards | Should include evidence of accuracy/precision with reference to international standards. Meets marketing, safety, quality management and environmental standards. |
| 2 | Child friendly device with all components must be non-hazardous and non-toxic | Device including all components and packaging is non-hazardous and non-toxic. The device must not have sharp edges, moving parts close to the body or rough surfaces |
| 3 | Affordable | Less than USD 150 |
| 4 | Meets UNICEF's children's rights and business principles | Adherence to UNICEF's Children's Rights and Business Principles |

**Fig 1. UNICEF target product profile criteria.**

synonyms and search terms for height/length measurement, children and devices. We also used Google searches to identify any height/length measurement devices that were commercially available at the time of searching. These were run in incognito mode, with location and trending searches turned off. These searches used various combinations of structured search terms and synonyms similar to the formal literature search, and we inspected the first 50 hits. Targeted searches were also conducted on manufacturer websites looking for additional manufacturing information on devices in the academic literature. Snowball searching was also done if any records that were identified appeared relevant [14].

Identified citations were collated using Endnote X8, before uploading to Covidence for de-duplication and screening. Two reviewers (TS and S Huang) independently assessed titles and abstracts of all retrieved citations. For potentially eligible studies, full texts were retrieved and assessed by two independent reviewers (TS and S Huang) according to the eligibility criteria. Disagreements during both stages were resolved either through discussion or consultation with a third reviewer (AW). We also screened all references of included studies to ensure that all eligible studies had been identified. We directly emailed manufacturers where further clarification on technical specification was required.

### Data collection, quality assessment and analysis

Data extraction was conducted using a customized spreadsheet, which we pre-tested and refined on four eligible studies. For each included study, we extracted data on the study design, country, setting and sample size, as well as the device characteristics, including weight, dimensions, method and ease of use, environmental durability, connectivity and power source. Additionally, quality assessment of all included studies was done using a Mixed Methods Appraisal Tool (MMAT) [15]. The MMAT, a tool comprised of two screening questions followed by five questions specific to study type. The MMAT was chosen as it allowed for standardized and objective appraisal of methodological quality across multiple different study types. Data extraction and quality assessment were performed by two independent reviewers, with differences resolved through discussion or involving a third reviewer. Findings from the scoping review were reported descriptively.

The UNICEF TPP outlines 34 key criteria across six domains (Fig 1). For each criterion, the TPP specifies an "ideal standard" and "minimum standard". Using all the information identified for each device, including the initial source as well as further searches for device manuals and other supplementary material, we scored each device on whether each of the 34 TPP criteria was met. Each criteria had a maximum score of two points—whether it met the ideal standard (score of 2), minimum standard (score of 1), or it was not met or we were unable to determine based on lack of information (score of 0). No additional weighting was used (i.e., all criteria had equal weighting). TPP terminology in some criteria such as "child friendly" were defined through consensus amongst the review team, informed by the UNICEF documentation. An overall score was then totaled for each device (i.e., maximum score of 68) and converted to a percentage. A hypothetical perfect device that met the ideal standard for all 34 TPP criteria would score 100%.

### Results

We identified 4,470 citations from database searches, and 876 duplicates were removed. Title and abstract screening of the remaining 3,594 unique citations identified 57 potentially eligible studies. After full text review, a total of 20 studies were included (Fig 2). The most common reasons for exclusion were measuring anthropometric outcomes other than height or length (15 studies), including adult populations only (6 studies) or using an ineligible intervention

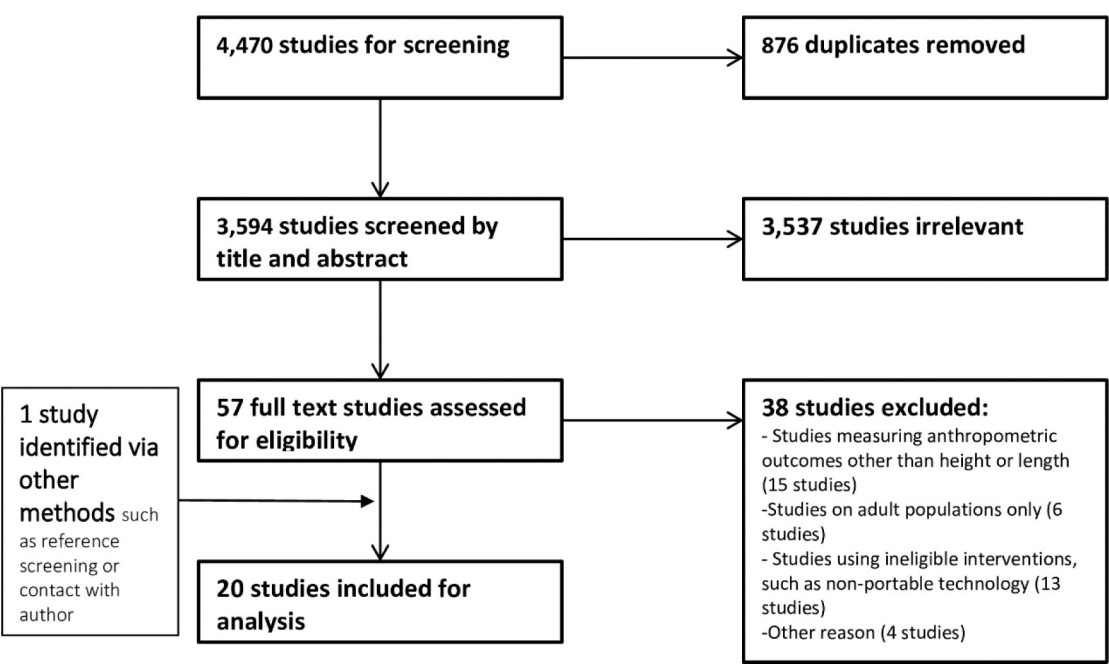

**Fig 2. PRISMA flowchart.**

(such as non-portable technology) (13 studies). Three eligible citations were conference abstracts, we attempted to contact the authors for further information without success, and hence used available data from the abstracts only. A number of devices were identified from grey literature searching and reference checks, however there lacked even minimal performance data for comparison against the UNICEF TPP, and hence were not included.

## Characteristics and quality of included studies

The 20 included studies were published between 1998 and 2022 (Table 1). Twelve studies (60%) were conducted in high-income countries (UK, Germany, USA, Sweden, Canada, Switzerland, Israel, Netherlands) and eight studies (40%) in low-and-middle-income countries (Guatemala, Kenya, Indonesia, China and South Sudan). Prospective validation studies were the most common study design (15 studies, 88%), reporting quantitative data only. These studies varied in quality, rigor, design and methodology. They were compared against an objective set of criteria designed by experts in reliability and agreement investigation in order to directly compare these studies (Appendix 3 in S1 Appendix) [16]. Two studies (12%) used a mixed-methods approach [17, 18]. The conference abstracts also used prospective validation designs. Twelve studies met 100% of the MMAT quality criteria, three studies met 80% of the quality criteria and two studies met only 40% of the quality criteria. We could not complete MMAT assessment for the conference abstracts (Appendix 4 in S1 Appendix).

Three studies (15%) included neonates only, two studies (10%) measured length of children <2 years, and three studies (15%) measured height of children >2 years. Twelve studies (60%) measured both height and length of children. Most studies (17 studies, 89%) used trained researchers only to perform measurements. In one study, two senior neonatologists performed the measurements [19], and in another study it was done by parents and team members, though it was not specified whether they received training [20]. Eleven studies (58%) described using formal device training prior to study commencement [1, 7, 17, 18, 21–27]. However,

**Table 1. Characteristics of included studies.**

| Author | Year of Publication | Country | Region | Country income level | Type of study | Population | Sample size | Setting | MMAT Score |
|---|---|---|---|---|---|---|---|---|---|
| Andrews et al [1] Kessing et al [25] | 2019 2020 | UK | Europe | HIC | Prospective validation study | Neonates <30 weeks gestational age | 17 | Inpatient medical facility | 80% |
| Bauman et al [21] Bauman et al [7] | 2015 2018 | Kenya | Africa | LMIC | Prospective validation study | Children 1 month– 8 years | 77 | Educational facilities OR outpatient medical facilities | 100% |
| Baxter et al [35] | 2015 | Canada | Americas | HIC | Prospective validation study | Children 0–18 years | 278 | Not specified | - |
| Blank et al [32] | 2017 | Sweden | Europe | HIC | Prospective validation study | Children aged 2–18 years | 62 | Hospital emergency department and inpatient facility | - |
| Bougma et all [26] | 2022 | Guatemala, Kenya, China | Africa, Asia and Americas | LMIC | Prospective validation study | Children 0–59 months | 600 | Health surveillance system teams | 100% |
| Conkle et al [17, 22] | 2018 2019 | USA | Americas | HIC | Prospective validation study + mixed methods | Children under 5 years | 474 | Day care OR medical facility OR religious facility | 100% |
| Glock et al [33, 34] | 1999 | Germany | Europe | HIC | Prospective validation study | Children (age not specified) | 101 | Medical inpatient and outpatient facility | 40% |
| Jefferds et al [18] | 2022 | Guatemala, Kenya, China | Africa, Asia and Americas | LMIC | Mixed methods | Children 0–59 months | 1258 caregivers 29 experts & assistants | Health surveillance system teams | 100% |
| Leidman et al [27] | 2022 | South Sudan | Africa | LMIC | Prospective validation study | 6–59 months | 539 | Health surveillance system teams | 100% |
| Mayol-Kreiser, et al [29] | 2015 | USA | Americas | HIC | Prospective validation study | Children and adults aged 3–80 years | 128(25)[a] | Not specified | 100% |
| Orozco et al [24] | 2015 | Guatemala | Americas | LMIC | Prospective validation study | Neonates | 29 | Hospital inpatient facility | - |
| Penders et al [23] | 2015 | Netherlands | Europe | HIC | Prospective validation study | Children and adolescents aged 2–18 years | 54 | Hospital outpatient facility | 100% |
| Sokolover et al [19] | 2014 | Israel | Middle East | HIC | Prospective validation study | Neonates | 54 | Hospital inpatient facility | 100% |
| Syafiq et al [31] Syafiq et al [30] | 2013 | Indonesia | Asia | LMIC | Prospective validation study | Children <2 years | 53 | Hospital outpatient facility | 100% |
| Watt et al [20] | 1998 | UK | Europe | HIC | Prospective validation study | Children (age not specified) | 32 | Medical outpatient facility | 40% |
| Wetzel et al [28] | 2018 | Switzerland | Europe | HIC | Prospective validation study | Children 0–13 years | 627 | Hospital emergency department | 80% |

HIC = high-income country; LMIC = low-middle income country; MMAT = mixed-methods assessment tool

[a] 25 individuals in this sample were children <12 years of age

only three studies specified the duration or quality of training [18, 26, 27]. These studies all ran multi-day detailed training for all members of the study team.

In five studies (25%) measurements were performed by a single team member [17, 28–31]. In five studies (25%) two members of the study team worked together to measure one individual participant [7, 22]. In three studies (15%) the participant cohort was split in two, with a

proportion having a single measurer and the other proportion having measurements performed by two or more team members [20, 21, 23]. A further two studies (10%) used two research staff to perform manual measurements but only one staff member for digital measurements [19, 24]. Five studies (25%) did not specify the number of team members used to measure a participant [1, 25, 32–35]. Five studies (25%) specifically rotated the order of measurements between the standard and digital devices, four studies (20%) did not rotate and eleven studies (55%) did not specify.

## Characteristics of devices and assessment against UNICEF target product profile

Twelve height/length measurement devices were described across the 20 studies (Table 2). These devices included; SCANTIFY, Leica Disto, AutoAnthro, Gulliver, GLM40, Cannon digital camera, unspecified digital camera, P2B2, Optisizer, Coolpix, GLM30 and PePA. Additional devices such as OneGrows and Kiko [36, 37] were found in the grey literature, however there was not an assessment of the device's reliability and precision so they were not included in the analysis. Eight devices weighed less than 1kg, one device weighed 6kg, and three devices did not have a weight reported. Six devices used digital camera technology (inclusive of 3D imaging), three used laser technology and three used ultrasound technology. Ultrasonic devices used ultrasonic transducers to send and receive ultrasound signals to measure distance. Laser devices worked by sending pulses of laser light, allowing calculation of time and therefore distance for the reflection to return. Cost ranged from approximately USD$45 to USD $2500. Contributing components to this cost included the device itself, software licenses and ancillary devices such as a tablet or smartphone. Notably, none of the devices that had been formally validated in the academic literature were currently commercially marketed explicitly for measuring height/length in people under the age of 18.

Quantitative data on accuracy were available for twelve devices. Nine devices reported this as the mean difference (mm) between a measurement using the digital device versus a standard device, usually a stadiometer or wooden height board. Four devices met the ideal standard for performance, described as a difference of 1mm or less. These were: Lecia DistoD2, Fuel 3D Scantify, AutoAnthro and a digital camera [1, 7, 17, 19, 22]. Two of these were measured in a controlled environment [19, 22] and two were measured in the field [1, 7]. However, when AutoAnthro was retested in field, outside of a controlled environment, the mean difference ranged from 30-620mm, and therefore it no longer met the UNICEF TPP criteria in this context. One device, the BOSCH GLM30, met the UNICEF TPP criteria for minimum standard, reporting a mean difference of 3mm [32]. The remaining four devices, Gulliver G-100 and three non-specified digital cameras reported mean differences ranging from 3.5mm-28mm, not meeting the TPP criteria [20, 33, 34]. A further three devices had studies where Bland-Altman plots or other statistical analyses were used to assess reliability and validity [28, 30, 31]. The novel devices PePA, OptisizerApp and P2B2D were described as reliable and valid compared to standard measurements on the basis of these [28, 30, 35]. However, we could not directly compare these metrics to the UNICEF TPP standards. Kiko, OneGrows and other devices from the grey literature did not provide a formal accuracy assessments [36, 37].

Quantitative data on precision was available for six devices. Only one device, AutoAnthro when tested in the field, reported precision as the mean difference (mm) between measurements using the same device. In this case, the mean difference of 40-77mm did not uphold the ideal UNICEF TPP criteria [26, 27]. The other five devices (Leica Disto D2, Bosch GLM40, P2B2D, cannon digital camera and unspecified digital cameras) as well as AutoAnthro when tested in a controlled environment documented precision using technical errors of

Table 2. Device characteristics.

| Device | Technology used | Weight | Dimensions | Tested for recumbent or standing use | Accuracy* | Precision** | Connectivity | Power requirement | Digital display | Training required? | Time needed (sec) | Price (USD) |
|---|---|---|---|---|---|---|---|---|---|---|---|---|
| PePA# [35] | Laser | - | - | Recumbent and standing | - | - | - | - | - | - | 8.4 | - |
| SCANIFY [1, 25] | Camera | 0.544kg | 25x24x3cm | Recumbent | 1mm | - | USB | Mains power connection needed | - | Yes | 0.1 | 1490 for device + cost of phone/computer to transfer data to (variable). Software is free with the device. |
| Leica Disto D2 [7, 21, 41] | Laser | 0.1kg | Headpiece: 3x 36 cm Leica meter: 11x4x2 cm | Recumbent and standing | 1mm | - | - | Battery operated | Yes | Yes | - | 179 for device |
| AutoAnthro System^ [17, 22] | Camera (3D imaging) + smartphone/iPad | <0.5kg camera + 0.1–0.5kg smart phone/iPad | - | Recumbent and standing | 1mm | 77mm | WIFI | USB rechargeable | Yes | Yes | 68 | 379 for AutoAnthro software + 300–600 for camera + cost of iPad/phone (dependent on model approx. 800) |
| AutoAnthro# [26, 27] | Camera (3D imaging) + smart phone/iPad | 0.1–0.5kg camera + 0.1–0.5kg smart phone or iPad | Camera: (version 1: 109 x18 x 24 mm, version 2: 61 mm x 26mm + dimensions of smart phone/iPad | Recumbent and standing | 30-620mm | 40-70mm | WIFI | USB rechargeable | Yes | Yes | <1 | 379 for AutoAnthro software + 300–600 for camera + cost of iPad/phone (dependent on model approx. 800) |
| Gulliver G-100 [34] | Ultrasound | - | - | Standing | 28mm | - | - | - | - | - | - | 665 for device |
| BOSCH GLM40 [29, 42] | Laser | 0.527kg | Plate11x27.7cm Laser device 5x4.1x2.4cm | Standing | 3.5mm | - | - | AAA batteries | Yes | Yes | <0.5 | 74 for device |
| Canon digital camera [23] | Camera | 0.755kg | 13.9 x 10.4 x 7.9cm | Standing | 5mm | - | USB and WIFI | Rechargeable battery | - | Yes | - | 1427 for software/camera system + computer cost (variable approx. 1000) |
| Digital camera [19] | Camera | - | - | Recumbent | 0.2mm | - | USB | USB rechargeable | - | Yes | - | - |

(Continued)

**Table 2.** (Continued)

| Device | Technology used | Weight | Dimensions | Tested for recumbent or standing use | Accuracy* | Precision** | Connectivity | Power requirement | Digital display | Training required? | Time needed (sec) | Price (USD) |
|---|---|---|---|---|---|---|---|---|---|---|---|---|
| P2B2D Prototype- [30, 31] | Ultrasound | 6kg | - | Recumbent | - | - | - | - | - | Yes | - | - |
| Optisizer App [28] | Camera | 0.185kg | 14.4 x 7.7 x 1.3 cm (and 20×20 cm sized tag) | Recumbent and standing | - | - | WIFI | USB rechargeable | Yes | - | <30 | App cost not specified + cost of iPhone/android (variable) |
| Nikon Coolpix S-3300 camera [24] | Camera | 0.128kg | 9.5 x 5.8 x 1.9 cm | Recumbent | 22mm | - | USB | USB rechargeable | - | - | - | - |
| BOSCH GLM 30 [32] | Laser | 0.09kg | 10.5 x 4.1 x 2.4cm | Recumbent and standing | 3mm | - | - | AAA batteries | Yes | Yes | <0.5 | 74 for device |

a Price at time of literature searching, or otherwise based on data sources identified in the review

b not otherwise specified

*mean difference calculated in mm between new device tested and current gold standard measurement device specified in paper

** mean difference calculated in mm between individual measurements (taken by either the same or different users) using the same device

# AutoAnthro version one when tested in a controlled environment

^ AutoAnthro version two and three when tested in field

# Pediatric Platform for Anthropometry

measurement (TEM) and Bland-Altman plots [7, 19, 22, 23, 29, 31]. Each of these studies reported the precision of the device to be comparable to that of manual measurements. However, we could not directly compare these metrics to the UNICEF TPP standards. Kiko, One-Grows and other devices from the grey literature did not provide a formal precision assessments [36, 37].

Twelve devices were scored against the 34 criteria of the UNICEF TPP (Appendix 2 in S1 Appendix). Available studies focused mainly on device accuracy, and lacked information on most other TPP criteria. The mean number of TPP criteria assessed was only 10 of the 34. Table 3 illustrates that of the criteria that were able to be assessed, Leica DistoD2 scored highest (41%) on the TPP matching analysis, followed by AutoAnthro in a controlled environment (33%) and GLM30 (32%).

Two studies assessed user experiences [17, 18], both of which looked at AutoAnthro. The studies found that it was at least as acceptable to capture measurements via scan technology as it was with manual measurement, with a ffavorable perception of AutoAnthro predominant, particularly among anthropometrists. Benefits included a more streamlined process, no need for other resources such as recording documents, being simple to learn and time saving. However, there were limitations such as needing a controlled environment and to remove large items of clothing which was particularly problematic in cold countries. Scanning was also slower and less dependable with an uncooperative child due to the requirement for them to stay still for the duration of the scan. This is a potentially high number of children, as it was noted that 30–50% of children were bothered by having height measured.

## Discussion

This scoping review identified 12 portable, digital devices for child height/length measurement. However, there are insufficient data publicly available on these devices, and we cannot determine conclusively whether they meet UNICEF's criteria for the hypothetical ideal device that meets clinical and public health needs. Some devices show promise as they are lightweight, accurate, digital and rechargeable–all of which are key advantages for real-world use. These devices that are accurate, lightweight and digital show high potential for commercialisation and use within clinical and public health settings. The grey literature search highlighted that portable, digital devices are entering the commercial market. Devices such as Kiko and One-Grows are commercially available for child height/length measurement, however there are no performance data available, particularly in relation to their accuracy and precision. More data on all key criteria outlined by the UNICEF TPP would be required before they can be considered for use in formal health programs. Despite this, the existence of these devices suggests that there is not only a need, but perhaps also a demand from individuals and families. None of the devices that have been validated in the academic literature are marketed and available for use for height/length measurement of children.

Many of the devices in the literature were assessed in a controlled environment. It cannot be assumed that their findings in such settings would translate to accurate use in the field, particularly in low- and middle-income countries. For example, while the AutoAnthro device was accurate and precise in a controlled environment, it did not perform well in clinical contexts in LMICs. Real-world factors such as temperature, lighting, power point connections, clothing specifications for scans and child co-operation need to be accounted for in the design of these products. Until these devices are designed with these operational considerations in mind, they are unlikely to succeed on a larger scale in diverse contexts. Furthermore, current cost of digital portable devices is significantly higher than a stadiometer. Additionally, operational

**Table 3. Summary of devices scored against the UNICEF target product profile.**

| Domain | Number of criteria (maximum score)* | Scantify [1, 25] Score (%) | Leica Disto D2 [7, 21] Score (%) | AutoAnthro* controlled environment [17, 22] Score (%) | Autoanthro* in field [26, 27] Score (%) | Gulliver [34] Score (%) | GLM40 [29] Score (%) | Cannon Digital Camera [23] Score (%) | Digital Camera [19] Score (%) | P2B2 [30] Score (%) | Optisizer [28] Score (%) | Coolpix [4] Score (%) | GLM30 [32] Score (%) | PePA [35] Score (%) |
|---|---|---|---|---|---|---|---|---|---|---|---|---|---|---|
| Operational & functional requirements | 11 (22) | 5/22 (23%) | 10/22 (45%) | 10/22 (45%) | 10/22 (45%) | 5/22 (23%) | 7/22 (32%) | 7/22 (32%) | 7/22 (32%) | 5/22 (23%) | 8/22 (36%) | 7/22 (32%) | 8/22 (36%) | 6/22 (27%) |
| Performance requirements | 5 (10) | 6/10 (60%) | 4/10 (40%) | 4/10 (40%) | 2/10 (20%) | 0/10 (0%) | 2/10 (20%) | 0/10 (0%) | 2/10 (20%) | 0/10 (0%) | 0/10 (0%) | 0/10 (0%) | 4/10 (40%) | 3/10 (30%) |
| Product requirements | 6 (12) | 0/12 (0%) | 4/12 (33%) | 0/12 (0%) | 0/12 (0%) | 0/12 (0%) | 0/12 (0%) | 0/12 (0%) | 0/12 (0%) | 0/12 (0%) | 0/12 (0%) | 0/12 (0%) | 0/12 (0%) | 0/12 (0%) |
| User requirements | 6 (12) | 8/12 (66%) | 8/12 (67%) | 9/12 (75%) | 7/12 (58%) | 0/12 (0%) | 8/12 (67%) | 4/12 (33%) | 5/12 (42%) | 5/12 (42%) | 6/12 (50%) | 6/12 (50%) | 8/12 (67%) | 0/12 (0%) |
| Supply chain requirements | 2 (4) | 0/4 (0%) | 0/4 (0%) | 0/4 (0%) | 0/4 (0%) | 0/4 (0%) | 0/4 (0%) | 0/4 (0%) | 0/4 (0%) | 0/4 (0%) | 0/4 (0%) | 0/4 (0%) | 0/4 (0%) | 0/4 (0%) |
| Commercialization Requirements | 4 (8) | 0/8 (0%) | 2/8 (25%) | 0/8 (0%) | 0/8 (0%) | 0/8 (0%) | 2/8 (25%) | 0/8 (0%) | 0/8 (0%) | 0/8 (0%) | 0/8 (0%) | 0/8 (0%) | 2/8 (25%) | 0/8 (0%) |
| **All domains** | **34 (68)** | **19/68 (28%)** | **28/68 (41%)** | **23/68 (33%)** | **19/68 (28%)** | **5/68 (7%)** | **19/68 (28%)** | **11/68 (16%)** | **14/68 (21%)** | **10/68 (15%)** | **14/68 (21%)** | **13/68 (19%)** | **22/68 (32%)** | **9/68 (13%)** |

* If the ideal standard is met for a criterion, a score of 2 is awarded. If the minimum standard is met for a criterion, a score of 1 is awarded. If the standard is not met or insufficient data are available to assess the standard, a score of 0 is awarded.

requirements of these conventional methods such as stadiometers not requiring a power source can be considered beneficial, particularly in LMICs.

UNICEF have identified that errors with manual height/length measurement using stadiometers occur and are problematic. These errors commonly result from parallax error (i.e. reading measurements from the wrong angle), difficulty reading measurements in poor lighting, and manual data entry errors [38]. These could all be avoided using digital technology. Given that reliable child growth data is essential to good public health programs, we consider digital devices–if accurate and reliable–to be potentially superior. However, current (or new) products require further development, particularly when considering in context use in LMICs. Currently there are no products that meet the complex requirements for in field use in LMICs, meaning stadiometers remain the gold standard.

Recent reviews looking at portable, digital measurement technology for neonates, infants and children have reported similar results. A systematic review by Van Gils et al looked at portable, digital technology for stress-free growth monitoring in preterm infants [39]. Like our review, they found that novel digital technologies show high promise, yet current devices need ongoing refinements and improvements in order to meet gold standard requirements, especially within clinical contexts. Similarly, a narrative review by Neale et al assessed recent innovations in portable height/length measurement devices [40], concluding that manual measurement devices such as measurement mats and portable stadiometers remain the preferred measuring tools until digital equivalents can be shown to be accurate and precise. Our study had a wider search strategy scope allowing for numerous new devices to be found compared to the other reviews. However, similar to these reviews, we identified a large gap in the current digital product market, despite devices showing promise none currently meet the UNICEF TPP criteria for an ideal product.

A strength of this study was the robust research process and methodology. This scoping review was conducted in accordance with a pre-specified protocol that was in line with contemporary methodological guidance [12]. We searched a wide range of sources, including grey literature, performed screening, data extraction and quality assessment in duplicate for quality control. We also performed the matching analysis using a publicly-available standard from a trusted international organization (UNICEF), which was also performed in duplicate to minimize errors. Despite this, some limitations must be acknowledged. During the TPP matching analysis, the reviewers did not test the discussed devices first hand, relying only on the literature for the analysis, limiting the assessment. Given these devices were not directly tested, it was difficulty to extrapolate from the studies the likely added difficulties in measuring length of young children lying down and how this affected device usage. The search strategy, in particular the choice of databases, provided a large scope addressing a wide array of literature. However, we must acknowledge that the not extending the search into other databases beyond Medline, Embase, CINAHL and Global Health means that certain devices could have been missed. Likewise, multiple different combinations of synonyms in our grey literature search found a diversity of results, however we must also acknowledge that reviewing only the first 50 hits would mean that some devices could have been missed. We were unable to locate additional information for two conference abstracts, meaning we could not fully assess the quality of these studies or their capacity to meet the TPP. It is possible that there is further information pertaining to the 12 identified devices that might have improved (or changed) their TPP matching scores, though these data may not be publicly available for commercial or intellectual property reasons. We tried to mitigate this possible bias through exhaustive searching and contacting manufacturers directly for additional information, though it is still possible that some relevant data were not obtained. The conclusions of the TPP matching analysis will likely change as new information or new devices are developed further. Additionally, it must be

noted that many studies were written prior to the TPP publication in 2017, making it plausible that these studies may not have considered some specifications desired by the TPP, impacting the devices overall score.

This review highlights a significant gap in both the current literature and commercial market for devices that measure height/length in children. The matching analysis demonstrates both the potential, as well as key areas for improvement, of current devices. Many devices currently available remain at the prototype or early development stage and have not been adapted for in field use or commercialization. We consider it unlikely that UNICEF's TPP will currently be fulfilled based on available devices. Despite this, some devices show high potential and reflection of their current status and the industry needs documented in the UNICEF TPP must be reviewed with further research. While further research investment on the most promising devices is warranted, development of novel devices or technologies remains worthwhile. Developers should transparently report the accuracy, precision and reliability of their devices, as well as clearly stating their operational, performance and user requirements. There is a clear impetus to accelerate the development and commercialization of growth measurement devices, to improve nutritional surveillance of children globally.

## Conclusions

This review identified a large gap in the academic literature and product market for portable digital growth measurement devices. Although some devices show promise further research is needed to test the validity of these devices in varying contexts. Continued development and commercialization will be important to improve reliability and validity of these devices for widespread use. Further studies are required to identify and develop devices that meet the UNICEF TPP criteria for height/length measurement devices.

## Supporting information

**S1 Checklist. Preferred Reporting Items for Systematic reviews and Meta-Analyses extension for Scoping Reviews (PRISMA-ScR) checklist.**
(DOCX)

**S1 Appendix. Contains Appendices 1–5.**
(DOCX)

## Acknowledgments

We gratefully acknowledge Lorena Romero who developed the search strategy.

## Author Contributions

**Conceptualization:** Tasmyn Soller, Joshua P. Vogel.

**Data curation:** Tasmyn Soller, Shan Huang, Alyce N. Wilson, Joshua P. Vogel.

**Formal analysis:** Tasmyn Soller, Shan Huang, Alyce N. Wilson, Joshua P. Vogel.

**Investigation:** Tasmyn Soller, Shan Huang, Joshua P. Vogel.

**Methodology:** Tasmyn Soller, Shan Huang, Alyce N. Wilson, Joshua P. Vogel.

**Project administration:** Tasmyn Soller, Joshua P. Vogel.

**Supervision:** Alyce N. Wilson, Joshua P. Vogel.

**Writing – original draft:** Tasmyn Soller.

**Writing – review & editing:** Tasmyn Soller, Shan Huang, Sayaka Horiuchi, Alyce N. Wilson, Joshua P. Vogel.

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
