## [Decision Letter · Decision Letter 0]

12 Jan 2023

PONE-D-22-30401

Portable digital devices for child height and length measurement: a scoping review and target product profile matching analysis

PLOS ONE

Dear Dr. Soller,

Thank you for submitting your manuscript to PLOS ONE. After careful consideration, we feel that it has merit but does not fully meet PLOS ONE’s publication criteria as it currently stands. Therefore, we invite you to submit a revised version of the manuscript that addresses the points raised during the review process.

We look forward to receiving your revised manuscript.

Kind regards,

Alex Schaefer, PhD

Associate Editor

PLOS ONE

Journal Requirements:

2. Please complete a PRISMA-ScR checklist (available at https://www.equator-network.org/wp-content/uploads/2018/09/PRISMA-ScR-Fillable-Checklist-1.docx) and upload it as supplementary file.

  "This research received no specific grant from any funding agency in the public, commercial or not-for-profit sectors. JPV is supported by a NHMRC Investigator Grant. All authors declare: no support from any organisation for the submitted work; no financial relationships with any organisations that might have an interest in the submitted work; no other relationships or activities that could appear to have influenced the submitted work."

4. Please ensure that you include a title page within your main document. You should list all authors and all affiliations as per our author instructions and clearly indicate the corresponding author.

Additional Editor Comments:

The reviewers feel that the literature search of this review needs to be updated since the search performed in the current version of the manuscript was done almost one year ago. The reviewers also note that the methodology and discussion need revisions. 

Reviewers' comments:

Reviewer's Responses to Questions

**Comments to the Author**

1. Is the manuscript technically sound, and do the data support the conclusions?

Reviewer #1: Yes

Reviewer #2: No

Reviewer #3: Partly

2. Has the statistical analysis been performed appropriately and rigorously? 

Reviewer #1: Yes

Reviewer #2: N/A

Reviewer #3: Yes

3. Have the authors made all data underlying the findings in their manuscript fully available?

Reviewer #1: Yes

Reviewer #2: Yes

Reviewer #3: Yes

4. Is the manuscript presented in an intelligible fashion and written in standard English?

Reviewer #1: Yes

Reviewer #2: Yes

Reviewer #3: Yes

5. Review Comments to the Author

Reviewer #1: This is a timely, interesting, and well written paper. Please check text for minor typos (e.g., line 109, the should be they).

Going through the list of devices outlined in Table 2, most seem to be repurposed digital/laser measuring devices. Some were discontinued and no longer available. A quick search indicates that a person or object's height can be measured with an iPhone. I am not an expert in this specific technology, but I do work in digital app development and my sense is that the field today is far ahead of the technology outlined in the table. Free or low-cost smartphone apps are now available for quantifying numerous body dimensions, although perhaps such software has not been designed for use in young children; but that would be trivial advance.

While I think the authors have done a good job reviewing the available literature, am I right that the digital world has or can move far beyond repurposed digital tape measures for evaluating children's length or height?

Reviewer #2: This manuscript is an important contribution to the overall quality of the children's anthropometric data collection. However, we would like to suggest some major revisions:

1. Expend the scope of the literature review up to Dec 2022. Some important articles were recently published and are not included in this review as the authors limited their search to almost 1 year ago (18 January 2022). Few examples:

Bougma et al., American Journal of Clinical Nutrition. 2022 Jul 6;116(1):97-110. doi: 10.1093/ajcn/nqac064.

Jefferds et al., Current Development in Nutrition. 2022 Apr 19;6(6). doi: 10.1093/cdn/nzac085.

Leidman et al., JMIR Biomed Eng 2022;7(2): e40066 doi:10.2196/40066

2. On the performance requirement of the UNICEF TPP, lease have separate section of precision besides the accuracy section included. Include a precision column in Table 2. Also add the term accuracy to the * for mean difference

3. Please nuance the main conclusion about some devices meeting the criteria for accuracy as most were validated in controlled setting. Please refer to the suggested articles in point one. The performance was not consistent in other settings

4. Cost of devices: Please consider the cost of acquisition of the data for the devices not providing directly/immediately the results. See the example of Auto Anthro. Also, considerer the cost recurrent licensing.

5. Performance requirements: please provide details on each device regarding "Immediate results"

6. References: Please review and reformat all the reference section

Minor comments:

Introduction

Line 89: include length-for-age also in the definition of stunting, not only height-for-age.

Line 91: include weight-for-length in the definition of wasting, not only weight-for-height

L91: I am not sure that obesity as defined by weight-for age z-score is a ration.

L110: Could you specify what you mean by "inconvenient' to use

Methods:

Line 148: typo. Please proofread all the manuscript

L163: See major comment. Update the scope of the literature review

L244: typo

Figure 1: please specify "screened by title and abstract

Results:

- See major comment for inclusion of separate section on precision.

- Please be specific on the terms used. Clarify what performance is related to ... accuracy or precision or both and be specific when talking about accuracy versus precision. Do not pull them together under terms like performance.

Reviewer #3: Overall comments

This study aimed to conduct a scoping review to identify and evaluate portable digital devices that are currently available to assess length and height measurement in children and adolescents <18 years of age. This is an interesting study and informative to understand the current landscape of digital portable devices available for anthropometric measurements. However, there are several key components of the methods are unclear and the interpretation of the findings does not take into account that the evidence from low- and middle-income countries is very little, which warrant discussion and tempering the conclusion in the abstract. I have made several comments below to strengthen the clarity of the methods and interpretation of the results.

Major comments

Introduction

o Please clarify how stadiometers or length boards are considered “inconvenient to use”. Many in the field, particularly those working in low- and middle-income country context would consider stadiometers and length boards are the most pragmatic tools to measure height and length. Clarity around how these tools are inconvenient and how digital tools might be 'simple to use' in any given context with varying levels experience with technological devises among users would be important.

Methods

Study design – please provide a table listing the UNICEF TPP criteria within the study design component of the methods as this is a key component of this study methodology and design. The study design without the specificity of what these 34 criteria are appears incomplete to the reader.

Literature search –

I am surprised that some of the large and interdisciplinary databases, e.g., Pubmed, Web of Science, and Scopus – were not searched as these databases are likely to index most of the public health and clinical literature relevant to this review. Would the authors please clarify the rationale? Including such a rationale in the limitations section, I expect, would be warranted.

Please cite Appendix 2 in the text to clarify that the search terms are detailed in the supplementary material.

o Lines 186-187: Please provide additional details regarding the Mixed Methods Appraisal Tool for clarity. It is unclear what this tool entails and how it is similar or different from other tools of quality assessment.

o Line 191: Please provide the UNICEF TPP Box 1 earlier, in the study design section for clarify.

o Please include either pragmatic definitions of how each criteria in UNICEF TPP Box 1 was operationalized for scoring or use-case examples for scoring each component. It is unclear from the current description, for example, how “long storage life” or “long operational life” or “operates across wide ranging climatic conditions” were systematically scored.

Results:

o Lines 220-221: Please provide additional details regarding the validation designs used in each study. The types of validations and the rigor of validation can vary substantially between studies, therefore such detail will be useful to interpret the utility of these tools. I recommend a table format for clarity (I see that further details are provided in the subsequent two paragraphs generalizing the validation exercises under taken but it is not clear to me which device validation was completed with which design).

o Please also provide additional details regarding the results for the MMAT score – a table in the appendix score each study against the MMAT criteria will be important for clarity.

Discussion:

o Majority of the studies found were developed and validate for use in high-income countries, and only 5 studies, tested in only 3 low- and middle-income countries, that tested the validity of digital portable anthropometry tools. This is a major gap and the very little evidence and development of tools in LMICs suggest that much work is needed here and that the technological barriers post by digital tools, which were not necessarily created with LMIC contexts in mind, certainly do not outweigh the benefits of tools such as the stadiometer. This warrants further comment in the discussion section (i.e. the utility of such tools that have very minimal evidence or validation in LMIC).

Abstract

o I find the conclusion of the abstract is overstated as the three devises – Leica DistoD2, Authoanthro, and GLM30 – all fail at least one of the UNICEF TPP minimum criteria. As a result, I suggest that no device names be used in the conclusion of the abstract. Rather, I suggest a more balanced and cautious interpretation of the data in the last sentence as follows: “Although some devices show promise, further research is needed to test the validity of these devices in varying contexts, and continued development and commercialization will be important to improve reliability of these devices for widespread use.”

Minor comments

Introduction

- Please use the definitions of <-2 standard deviations from the reference median when describing stunting and wasting for accuracy and clarity (rather than “low”).

- Eligibility criteria section: please clarify why the year 1992 was chosen specifically as the start year for searching for relevant articles.

6. PLOS authors have the option to publish the peer review history of their article (what does this mean?). If published, this will include your full peer review and any attached files.

Reviewer #1: No

Reviewer #2: **Yes: **Karim Bougma

Reviewer #3: No

---

## [Author Response · Author response to Decision Letter 0]

1 Mar 2023

As per the document 'response to reviewers'

Response: We have amended the manuscript to meet attached style guidelines 

2. Please complete a PRISMA-ScR checklist (available at https://www.equator-network.org/wp-content/uploads/2018/09/PRISMA-ScR-Fillable-Checklist-1.docx) and upload it as supplementary file.

Response: We have attached the completed checklist as a supplementary file 

 "This research received no specific grant from any funding agency in the public, commercial or not-for-profit sectors. JPV is supported by a NHMRC Investigator Grant. All authors declare: no support from any organisation for the submitted work; no financial relationships with any organisations that might have an interest in the submitted work; no other relationships or activities that could appear to have influenced the submitted work."

Response: Amended statement as follows

This research received no specific grant, sources of funding or material support from any agency in the public, commercial or not-for-profit sectors. The research was supported by the Burnet Institute, however there was no funding received from this institution. JPV is supported by a NHMRC Investigator Grant. Funders had no role in study design, data collection and analysis, decision to publish, or preparation of the manuscript. All authors declare: no support from any organisation for the submitted work; no financial relationships with any organisations that might have an interest in the submitted work; no other relationships or activities that could appear to have influenced the submitted work. 

4. Please ensure that you include a title page within your main document. You should list all authors and all affiliations as per our author instructions and clearly indicate the corresponding author.

Response: We have amended the manuscript accordingly

Response: We have amended the manuscript accordingly

Additional Editor Comments:

The reviewers feel that the literature search of this review needs to be updated since the search performed in the current version of the manuscript was done almost one year ago. The reviewers also note that the methodology and discussion need revisions. 

Response: Thank you for your advice regarding expanding the scope of the literature review. We re ran the search on the 2nd Feb 2023. We have updated the results and discussion accordingly. There were three additional quality papers added. These provided valuable data on the use of portable height/length measurement devices infield, especially in low- and middle-income countries. One of these studies also added depth though including analysis of user perceptions/experiences when using these devices. 

One additional systematic review was also found through the search. This was van Gils, R. H. J., Wauben, L. S. G. L., & Helder, O. K. (2022). Body size measuring techniques enabling stress-free growth monitoring of extreme preterm infants inside incubators: A systematic review. PloS one, 17(4), e0267285. https://doi.org/10.1371/journal.pone.0267285. While this was not directly included in the analysis, the review provided a valuable reference point for comparison and analysis in our discussion. The comparatively significant number of papers published on this topic in 2022 not only enhances our scoping review but also suggests that now is a pertinent time to publish such as review as research into this topic is accelerating. 

Please see direct responses to reviewers’ comments about methodology and discussion below. 

Reviewers' comments:

Reviewer's Responses to Questions

Comments to the Author

1. Is the manuscript technically sound, and do the data support the conclusions?

Reviewer #1: Yes

Reviewer #2: No

Reviewer #3: Partly

Response: Noted, thanks. Please see direct response to comments/revisions. 

2. Has the statistical analysis been performed appropriately and rigorously?

Reviewer #1: Yes

Reviewer #2: N/A

Reviewer #3: Yes

Response: Noted, thanks.

3. Have the authors made all data underlying the findings in their manuscript fully available?

Reviewer #1: Yes

Reviewer #2: Yes

Reviewer #3: Yes

Response: Noted, thanks.

4. Is the manuscript presented in an intelligible fashion and written in standard English?

Reviewer #1: Yes

Reviewer #2: Yes

Reviewer #3: Yes

Response: Noted, thanks.

5. Review Comments to the Author

Reviewer #1: This is a timely, interesting, and well written paper. Please check text for minor typos (e.g., line 109, the should be they).

Going through the list of devices outlined in Table 2, most seem to be repurposed digital/laser measuring devices. Some were discontinued and no longer available. A quick search indicates that a person or object's height can be measured with an iPhone. I am not an expert in this specific technology, but I do work in digital app development and my sense is that the field today is far ahead of the technology outlined in the table. Free or low-cost smartphone apps are now available for quantifying numerous body dimensions, although perhaps such software has not been designed for use in young children; but that would be trivial advance.

While I think the authors have done a good job reviewing the available literature, am I right that the digital world has or can move far beyond repurposed digital tape measures for evaluating children's length or height?

Response: Thank you for taking the time to review this manuscript. It is really valuable to have your perspective as an app developer. As a team of multidisciplinary health care professionals and researchers the perspective you are able to provide is extremely valuable to the translatability and usefulness of this review to industry and product development. 

In response to your comments:

We have reviewed for minor typos and corrected the typo on line 109 accordingly. 

We also found a number of devices in the grey literatures including apps which could be used to measure body dimensions. However, none of these devices were formally validated to ensure accuracy and precision when used in the paediatric population. While we are not app developers, we agree that this must be a trivial advance to current technology. UNICEF has highlighted the need for technological innovation in height measurement since the target product profile for such devices was released in 2017. We hope this review will highlight this gap, creating an impetus to accelerate advances and promote developers to translate technical improvements seen in other areas to improve measurement and nutritional surveillance of children globally.

Reviewer #2: This manuscript is an important contribution to the overall quality of the children's anthropometric data collection. However, we would like to suggest some major revisions:

Thank you for taking the time to review this manuscript. As an anthropometry specialist with decades of experience your insights and opinions are extremely valuable. You have provided very sound feedback that we feel has greatly enhanced and strengthened our review, so thank you for all the thought and consideration you put into writing it. We have addressed individual comments below. 

1. Expend the scope of the literature review up to Dec 2022. Some important articles were recently published and are not included in this review as the authors limited their search to almost 1 year ago (18 January 2022). Few examples:

Bougma et al., American Journal of Clinical Nutrition. 2022 Jul 6;116(1):97-110. doi: 10.1093/ajcn/nqac064.

Jefferds et al., Current Development in Nutrition. 2022 Apr 19;6(6). doi: 10.1093/cdn/nzac085.

Leidman et al., JMIR Biomed Eng 2022;7(2): e40066 doi:10.2196/40066

Response: Thank you for your advice regarding expanding the scope of the literature review. We re ran the search on the 2nd Feb 2023. We have updated the results and discussion accordingly. In this search we have included the papers suggested above. These provided valuable data on the use of portable height/length measurement devices infield, especially in low- and middle-income countries (LMIC). This in field research is particularly important to ensure that the academic literature is translatable to real world clinical application. This was an area that was not thoroughly discussed prior to inclusion of these papers. Hence expanding the dates of this review added significant depth to the quality of our manuscript. One of these studies also added depth though including analysis of user perceptions/experiences, another important aspect to consider in order to allow for industry transition that was lacking in our prior analysis. 

One additional systematic review was also found through the search. This was van Gils, R. H. J., Wauben, L. S. G. L., & Helder, O. K. (2022). Body size measuring techniques enabling stress-free growth monitoring of extreme preterm infants inside incubators: A systematic review. PloS one, 17(4), e0267285. https://doi.org/10.1371/journal.pone.0267285. While this was not directly included in the analysis, the review provided a valuable reference point for comparison and analysis in our discussion. 

The comparatively significant number of papers published on this topic in 2022 not only enhances our scoping review but also suggests that now is a pertinent time to publish such a review as research into this topic has accelerated over the last 12 months. 

2. On the performance requirement of the UNICEF TPP, lease have separate section of precision besides the accuracy section included. Include a precision column in Table 2. Also add the term accuracy to the * for mean difference

Response: Thank you for addressing this. While we thoroughly addressed accuracy, we agree that precision is a very important aspect that was a significant gap in our earlier manuscript. 

We have included a paragraph on precision following the accuracy section of our results. We have copied it here for your reference:

Quantitative data on precision was available for six devices. Only one device, AutoAnthro when tested in the field, reported precision as the mean difference (mm) between measurements using the same device. In this case, the mean difference of 40-77mm did not uphold the ideal UNICEF TPP criteria (23, 24). The other five devices (Leica Disto D2, Bosch GLM40, P2B2D, cannon digital camera and unspecified digital cameras) as well as AutoAnthro when tested in a controlled environment documented precision using technical errors of measurement (TEM) and Bland-Altman plots (6, 16, 19, 20, 26, 28). Each of these studies reported the precision of the device to be comparable to that of manual measurements. However, we could not directly compare these metrics to the UNICEF TPP standards. Kiko, OneGrows and other devices from the grey literature did not provide a formal precision assessments (32, 33).

We have also updated Table 2 with a column on precision and added the term accuracy as specified above. We have also included precision as a key focus point in our discussion. 

3. Please nuance the main conclusion about some devices meeting the criteria for accuracy as most were validated in controlled setting. Please refer to the suggested articles in point one. The performance was not consistent in other settings

Response: Thank you for this advice, we have nuanced the conclusion accordingly. Key changes show that while some devices show promise it is important that devices are designed with purpose and validated within the context that they will be used.

We have copied the conclusion here for your reference:

This review identified a large gap in the academic literature and product market for portable digital growth measurement devices. Although some devices show promise further research is needed to test the validity of these devices in varying contexts. Continued development and commercialization will be important to improve reliability and validity of these devices for widespread use. Further studies are required to identify and develop devices that meet the UNICEF TPP criteria for height/length measurement devices. 

4. Cost of devices: Please consider the cost of acquisition of the data for the devices not providing directly/immediately the results. See the example of Auto Anthro. Also, considerer the cost recurrent licensing.

Response: Thank you for this suggestion, this is an important consideration when assessing the cost of such devices. We have updated the results, Table 2 and our discussion accordingly.

5. Performance requirements: please provide details on each device regarding "Immediate results"

Response: Thank you for prompting us to clarify this, we have updated Table 2 and our results accordingly. Each device (where data was available) now has a specific time for results listed in seconds. For reference we have defined immediate results as any result taking less than 1 second to process. 

6. References: Please review and reformat all the reference section

Response: Noted, and references have been reviewed and reformatted according to PLOS ONE guidelines. 

Minor comments:

Introduction

Line 89: include length-for-age also in the definition of stunting, not only height-for-age.

Response: Amended accordingly.

Line 91: include weight-for-length in the definition of wasting, not only weight-for-height

Response: Amended accordingly.

L91: I am not sure that obesity as defined by weight-for age z-score is a ration.

Response: We have amended the definition to match that published by the WHO.

Definition is copied here for reference. Obesity: a body mass index greater than 3 standard deviations above the WHO growth standard median. 

L110: Could you specify what you mean by "inconvenient' to use

Response: Thank you for allowing us to clear up ambiguity around this statement. We have recreated this paragraph to clarify perceived limitations with manual anthropometry. We have copied this paragraph here for your reference:

Measurement boards (stadiometers) are currently the gold-standard tool for obtaining child height and length, including in community-based child health surveillance activities (6). While they have been in widespread use for decades, they are cumbersome and heavy.(7) The process of manual data collection also increases the risk of transcription errors, negatively affecting data accuracy.

Methods:

Line 148: typo. Please proofread all the manuscript

Response: Amended accordingly. We have also proof read the manuscript for other typos. 

L163: See major comment. Update the scope of the literature review

Response: Please see response to major comment above. We have amended the dates in the methodology to match this revision. 

L244: typo

Response: Amended accordingly. We have also proof read the manuscript for other typos. 

Figure 1: please specify "screened by title and abstract

Response: Amended accordingly.

Results:

- See major comment for inclusion of separate section on precision.

Response: Please see response to major comment above. 

- Please be specific on the terms used. Clarify what performance is related to ... accuracy or precision or both and be specific when talking about accuracy versus precision. Do not pull them together under terms like performance.

Response: Thank you for flagging this important issue. By adding analysis on precision to our results and discussion we were able to highlight the importance of both accuracy and precision to the performance of height/length measurement devices. We also made sure we clearly differentiated between accuracy and precision, ensuring that they were no longer pooled into a single entity. 

Reviewer #3: Overall comments

This study aimed to conduct a scoping review to identify and evaluate portable digital devices that are currently available to assess length and height measurement in children and adolescents <18 years of age. This is an interesting study and informative to understand the current landscape of digital portable devices available for anthropometric measurements. However, there are several key components of the methods are unclear and the interpretation of the findings does not take into account that the evidence from low- and middle-income countries is very little, which warrant discussion and tempering the conclusion in the abstract. I have made several comments below to strengthen the clarity of the methods and interpretation of the results.

Thank you for taking the time to review this manuscript. You have provided well thought through feedback that we feel has greatly improved and strengthened our review. We found it particularly valuable reflecting on the translatability of these findings to LMICs, where accurate nutritional surveillance of children is paramount. We have addressed individual comments below. 

Major comments

Introduction

o Please clarify how stadiometers or length boards are considered “inconvenient to use”. Many in the field, particularly those working in low- and middle-income country context would consider stadiometers and length boards are the most pragmatic tools to measure height and length. Clarity around how these tools are inconvenient and how digital tools might be 'simple to use' in any given context with varying levels experience with technological devises among users would be important.

Response: Thank you for allowing us to clear up ambiguity around this statement. We have recreated this paragraph to clarify perceived limitations with manual anthropometry and possible advantages of digitalisation. We have copied this paragraph here for your reference:

Measurement boards (stadiometers) are currently the gold-standard tool for obtaining child height and length, including in community-based child health surveillance activities (6). While they have been in widespread use for decades, they are cumbersome and heavy.(7) The process of manual data collection also increases the risk of transcription errors, negatively affecting data accuracy. A digital, accurate measurement device would thus provide significant advantages for clinical assessment and nutritional surveillance of children. If such a device were portable, lightweight, and reliable, with simple data storage and transfer abilities, it would make child health assessments easier. This is especially pertinent in resource-limited settings, where the large, cumbersome measurement board can be particularly difficult to transport and use, and reliable data transcription and record-keeping can be challenging. 

Methods

Study design – please provide a table listing the UNICEF TPP criteria within the study design component of the methods as this is a key component of this study methodology and design. The study design without the specificity of what these 34 criteria are appears incomplete to the reader.

Response: Thank you for flagging this important issue. We have listed the UNICEF TPP criteria as Figure 1 in the study design section of the methodology. We have also greatly improved this figure by listing in detail a pragmatic definition of how each criterion was defined in our study. These definitions are based directly off the UNCIEF TPP. 

Literature search –

I am surprised that some of the large and interdisciplinary databases, e.g., Pubmed, Web of Science, and Scopus – were not searched as these databases are likely to index most of the public health and clinical literature relevant to this review. Would the authors please clarify the rationale? Including such a rationale in the limitations section, I expect, would be warranted.

Response: We agree that this is an important consideration which as a team we thoroughly reviewed before commencing our search. We decided on the use of Medline, Embase, CINAHL and Global Health databases following exploratory searches across multiple databases and consultation with an information specialist. We also thoroughly cross-checked the references of all papers and systemic or narrative reviews we found in order to ensure that we were not missing any important papers. While we feel that the search was thorough and allowed us to evaluate the most up to date literature on the topic, we do acknowledge the limitations provided by our choice of data bases which we have added to the limitations section of the discussion. 

Please cite Appendix 2 in the text to clarify that the search terms are detailed in the supplementary material.

Response: Amended accordingly.

o Lines 186-187: Please provide additional details regarding the Mixed Methods Appraisal Tool for clarity. It is unclear what this tool entails and how it is similar or different from other tools of quality assessment.

Response: We have added a sentence to clarify what the MMAT tool is. We have copied the sentence here for your reference. 

The MMAT, a tool comprised of two screening questions followed by five questions specific to study type. The MMAT was chosen as it allows for a standardized and objective appraisal of methodological quality across multiple different study types.

Further details as to how the MMAT tool is best utilised can be found in the MMAT user guide from MgGill University (reference 14): Hong QN. Mixed Methods Appraisal Tool (MMAT) Version 2018 User Guide2018 3rd April 2022 [cited 2022 5th May 2022]; 1:[11 p.]. Available from: http://mixedmethodsappraisaltoolpublic.pbworks.com/w/file/fetch/127916259/MMAT_2018_criteria-manual_2018-08-01_ENG.pdf.

o Line 191: Please provide the UNICEF TPP Box 1 earlier, in the study design section for clarify.

Response: Amended accordingly, please see comment above.

o Please include either pragmatic definitions of how each criteria in UNICEF TPP Box 1 was operationalized for scoring or use-case examples for scoring each component. It is unclear from the current description, for example, how “long storage life” or “long operational life” or “operates across wide ranging climatic conditions” were systematically scored.

Response: Amended accordingly, each criterion now has a pragmatic and practical definition based directly off the UNICEF TPP. Please see Figure 1 for details. 

Results:

o Lines 220-221: Please provide additional details regarding the validation designs used in each study. The types of validations and the rigor of validation can vary substantially between studies, therefore such detail will be useful to interpret the utility of these tools. I recommend a table format for clarity (I see that further details are provided in the subsequent two paragraphs generalizing the validation exercises under taken but it is not clear to me which device validation was completed with which design).

Response: We have created a table (Appendix 2) which directly compares all studies using validation designs. These were compared for rigor, methodolgy and quality using a methodological criterion created by experts in reliability and agreement investigation (reference for this methodology: Kottner J, Audigé L, Brorson S, Donner A, Gajewski BJ, Hróbjartsson A, et al. Guidelines for Reporting Reliability and Agreement Studies (GRRAS) were proposed. Journal of Clinical Epidemiology. 2011;64(1):96-106.)

o Please also provide additional details regarding the results for the MMAT score – a table in the appendix score each study against the MMAT criteria will be important for clarity.

Response: We have created a table (Appendix 3) with the results for each component of the MMAT score for each study. This should create clarity around results. 

Discussion:

o Majority of the studies found were developed and validate for use in high-income countries, and only 5 studies, tested in only 3 low- and middle-income countries, that tested the validity of digital portable anthropometry tools. This is a major gap and the very little evidence and development of tools in LMICs suggest that much work is needed here and that the technological barriers post by digital tools, which were not necessarily created with LMIC contexts in mind, certainly do not outweigh the benefits of tools such as the stadiometer. This warrants further comment in the discussion section (i.e. the utility of such tools that have very minimal evidence or validation in LMIC).

Response: Thank you for this insightful comment. Expanding the dates of the literature search to include an additional 3 papers looking at height/length measurement devices in LMIC contexts reinforces your view. These studies demonstrated that while devices such as AutoAnthro have been validated in controlled environments in high income countries this does not translate when tested in the field in LMICs. This review demonstrates that there is currently a significant design and knowledge gap for translational research within these contexts. This has been addressed in our ammedments to the discussion which we hope creates an impetus for future studies that focus on specific requirements for use in LMICs. 

We have coppied some ammendments here for your reference:

For example, while the AutoAnthro device was accurate and precise in a controlled environment, it did not perform well in a study in clinical contexts in LMICs. Real-world factors such as temperature, lighting, power point connections, clothing specifications for scans and child co-operation need to be accounted for in the design of these products. Until these devices are designed with these operational considerations in mind, they are unlikely to succeed on a larger scale in diverse contexts. 

…..

Given that reliable child growth data is essential to good public health programs, we consider digital devices – if accurate and reliable – to be potentially superior. However, current (or new) products require further development, particularly when considering in context use in LMICs. Currently there are no products that meet the complex requirements for in field use in LMICs, meaning stadiometers remain the gold standard. 

Abstract

o I find the conclusion of the abstract is overstated as the three devises – Leica DistoD2, Authoanthro, and GLM30 – all fail at least one of the UNICEF TPP minimum criteria. As a result, I suggest that no device names be used in the conclusion of the abstract. Rather, I suggest a more balanced and cautious interpretation of the data in the last sentence as follows: “Although some devices show promise, further research is needed to test the validity of these devices in varying contexts, and continued development and commercialization will be important to improve reliability of these devices for widespread use.”

Response: We have amended the conclusion of the study to be more balanced and cautious as you have outlined above. 

We have copied the new abstract conclusion here for your reference:

While 11 portable, digital devices exist for child height/length measurement, insufficient data are available to fully assess whether they meet the industry’s needs. Although some devices show promise, further research is needed to test the validity of these devices in varying contexts, and continued development and commercialization will be important to improve reliability of these devices for widespread use.

Minor comments

Introduction

- Please use the definitions of <-2 standard deviations from the reference median when describing stunting and wasting for accuracy and clarity (rather than “low”).

Response: We have amended this accordingly. All definitions are written as standard deviations from the reference median as per the WHO definition. 

We have copied the amendments here for your reference:

Stunting, defined as more than two standard deviations below the median height-for-age/length-for-age, affects approximately 162 million children under 5 years worldwide. 

……

Wasting is defined as more than two standard deviations below the median weight-for-height/weight-for-length and indicates recent and severe weight loss.

- Eligibility criteria section: please clarify why the year 1992 was chosen specifically as the start year for searching for relevant articles.

Response: Thank you for this question as we feel that this is an important consideration. As a team this is something we thought deeply about. We searched the literature from 1 January 1992 onwards, which captured 30 years of data. We did this as we believe that 30 years of data would allow for a thorough evaluation of all contemporary product as well as giving us insight into how this has changed over time. We acknowledge that most of the relevant data is from the last decade. However, feel that 30 years of data gave us a good idea of how this field has changed and evolved over time – providing a useful perspective for the review.

---

## [Decision Letter · Decision Letter 1]

30 May 2023

PONE-D-22-30401R1

Portable digital devices for child height and length measurement: a scoping review and target product profile matching analysis

PLOS ONE

Dear Dr. Soller,

Thank you for submitting your manuscript to PLOS ONE. After careful consideration, we feel that it has merit but does not fully meet PLOS ONE’s publication criteria as it currently stands. Therefore, we invite you to submit a revised version of the manuscript that addresses the points raised during the review process.

While Reviewer #1 has generally provided positive comments and finds the manuscript suitable for publication, Reviewer #4 raised several queries. I understand that it may not be possible to re-run the systematic search; however, I urge the authors to address the careful critiques provided by Reviewer #4 for greater clarity in the methods and results, and further reflections in the discussion, and revise the manuscript accordingly.   

We look forward to receiving your revised manuscript.

Kind regards,

Nandita Perumal, PhD

Guest Editor

PLOS ONE

Journal Requirements:

Reviewers' comments:

Reviewer's Responses to Questions

**Comments to the Author**

1. If the authors have adequately addressed your comments raised in a previous round of review and you feel that this manuscript is now acceptable for publication, you may indicate that here to bypass the “Comments to the Author” section, enter your conflict of interest statement in the “Confidential to Editor” section, and submit your "Accept" recommendation.

Reviewer #1: All comments have been addressed

Reviewer #4: (No Response)

2. Is the manuscript technically sound, and do the data support the conclusions?

Reviewer #1: Yes

Reviewer #4: Partly

3. Has the statistical analysis been performed appropriately and rigorously? 

Reviewer #1: Yes

Reviewer #4: N/A

4. Have the authors made all data underlying the findings in their manuscript fully available?

Reviewer #1: Yes

Reviewer #4: Yes

5. Is the manuscript presented in an intelligible fashion and written in standard English?

Reviewer #1: Yes

Reviewer #4: Yes

6. Review Comments to the Author

Reviewer #1: The authors have made all of my suggested revisions. I have no additional comments. This is a useful review on this topic.

Reviewer #4: Thank you for the opportunity to review this scoping review on different portable tools to measure child height and length. The current work describes studies identified through published and grey literature, and how they compare to the UNICEF TPP. Measuring child height and length objectively and with high validity is important, making this an interesting assessment. Enclosed, please find several queries to the authors with the intention of strengthening the manuscript.

Methods:

- The search was last conducted >1 year ago (Jan 2022). The researchers should consider re-running the search strategy.

- The search strategy is noted in Appendix, but this does not appear in Methods text. Reference to the corresponding Appendix should be added.

- From reviewing the search strategy, was infant or child length searched? It seems the preference was for height, although the term used for those <2 years is typically length, and child length is noted in the Title of the manuscript.

- In the grey literature searches (Google), why were the first 50 hits considered only? Is there a reference or methodology that supports this approach? Given that Google has strong algorithms that control what a searcher might observe, do the researchers believe this figure is appropriately representative of the tools available?

- Would it be possible to indicate which reviewers conducted the reviews and which reviewer was approached for consultation?

Results:

- Since the Abstract and Discussion note that 13 portable devices were identified, this should appear within the Results section. It can be determined by counting Table 2, but only that 16 studies were identified is mentioned in the Results. A sentence stating 13 portable devices would improve the manuscript clarity.

- To assess the UNICEFF TPP, it seems that several characteristics (e.g., cost) go beyond those mentioned as part of in the customized data extraction form. How were these characteristics captured?

- Had the researchers come across the following study in their searches?

Additional study:

Baxter, J.-A., Roth, D., Zlotkin, S., Yin, S., Milgram, P. and Mouzaki, M. (2015), Measuring Child Length and Height: Assessing the Accuracy of a Portable Infrared-based Digital Tool. The FASEB Journal, 29: 31.3. https://doi.org/10.1096/fasebj.29.1_supplement.31.3

- For completeness, the researchers should consider adding a simple table to the Appendix that lists the articles excluded and reasons for exclusion. These studies may be of interest to the readership.

Discussion:

- Line 284-85: Per above, it is noted that 13 portable devices were identified. This had not previously been noted in the Results, only the number of studies. Suggest adding number of unique devices within the results, as 16 studies is mentioned, and determining that there are 13 devices requires counting lines in the table.

- The Neale et al study is introduced in line 310, but then 2 sentences from line 311-16 (starting with UNICEF have identified...) appear before mention of Neale again. Are these 2 sentences supposed to appear here? They seem out of place. As well, being clear about how the methods/scope of the current review is different from Neale would improve the interpretability.

- Based on the UNICEF TPP findings and last few paragraphs of the Results, could the researchers provide further critical reflection in the Discussion about the practicality of tools available, if they were to be used in diverse settings? (i.e., LMICs) A good portion of space in the Introduction reflects on stunting and wasting, and which are conditions more likely to affect those in LMICs. Do these devices seem practical for use in LMICs? For example, could reflect on cost relative to existing standard tools (could look at seca or Harpenden for cost comparison) or power requirements (standard length and height boards do not currently require a power source).

- Could the researchers reflect further on the limitations of their own study methods as conducted?

(1) Given this is a review of the literature, the researchers likely didn't interact with any of the tools themselves. In other words, without ever seeing the tools, and just reading text, do the reviewers feel that they were adequately able to assess TPP characteristics? Comparing reports of some characteristics in the UNICEFF TPP could be subjective, especially given comparison is being made between studies.

(2) Measuring length and height is quite different, length is conventionally much more difficult to measure - largely related to the age of the child and their cooperation whilst being measured. There is information on length versus height in the Results, so this could be further reflected upon.

(3) As the UNICEF TPP was published in 2017 and several identified studies were published before this, it is quite plausible that they would not report on such characteristics. This could be reflected on further.

Appendix:

- The Tables presented in the Appendix do not align with the main manuscript (Table 1 in the Appendix is noted as Table 3 in text).

- Search strategy is provided in the Appendix, but not listed in the main paper. Within the Appendix, given the placement of mention of the search strategy, this should appear first (i.e., before UNICEF TPP)

- A table describing exclusions at full-text stage would strengthen the manuscript, and is standard practice for other review platforms

7. PLOS authors have the option to publish the peer review history of their article (what does this mean?). If published, this will include your full peer review and any attached files.

Reviewer #1: No

Reviewer #4: No

---

## [Author Response · Author response to Decision Letter 1]

21 Jun 2023

Dear Dr Perumal, 

Many thanks for your response. We appreciate the time and thought taken to rereview our manuscript. The reviewers have provided valuable feedback which we feel has strengthened our manuscript. 

Regards, 

Tasmyn, on behalf of the study team

Response to reviewers:

Reviewer #1: The authors have made all of my suggested revisions. I have no additional comments. This is a useful review on this topic.

Response: Thank you. 

Reviewer #4: Thank you for the opportunity to review this scoping review on different portable tools to measure child height and length. The current work describes studies identified through published and grey literature, and how they compare to the UNICEF TPP. Measuring child height and length objectively and with high validity is important, making this an interesting assessment. Enclosed, please find several queries to the authors with the intention of strengthening the manuscript.

Response: Thank you for your comments. Please find detailed responses below and accompanying revisions in the manuscript. 

Methods:

The search was last conducted >1 year ago (Jan 2022). The researchers should consider re-running the search strategy.

Response: 

As per the methods (Page 7), the search was re-run on the 2 February 2023 to ensure the most up-to-date papers were included. 

The search strategy is noted in Appendix, but this does not appear in Methods text. Reference to the corresponding Appendix should be added.

Response: we have now referenced the corresponding appendix in the methods.

From reviewing the search strategy, was infant or child length searched? It seems the preference was for height, although the term used for those <2 years is typically length, and child length is noted in the Title of the manuscript.

Response: Thank you for raising this, height and length terminology was comprehensively considered in the development of our search strategy. We are aware that height is measured standing in children >2 years and length is measured lying down for infants <2 years old and we carefully selected available terminology to reflect this in our search. This is demonstrated in our results whereby most studies measured length of children < 2 years, 5 of which exclusively measured length – suggesting this was adequately addressed in the search strategy. Please find these results described as follows: 

“Three studies (16%) included neonates only, two studies (11%) measured length of children <2 years, and three studies (16%) measured height of children >2 years. Eleven studies (58%) measured both height and length of children” 

In the grey literature searches (Google), why were the first 50 hits considered only? Is there a reference or methodology that supports this approach? Given that Google has strong algorithms that control what a searcher might observe, do the researchers believe this figure is appropriately representative of the tools available?

Response: Thank you for raising this, we acknowledge that searching google can be overwhelming due to the abundance of information. We are also aware that google search engines use relevancy rankings to bring the most pertinent results to the top. 

Our search was run in incognito mode with location and trending searches turned off. Our methodology was initially tested using a number pilot run whereby multiple team members searched the grey literature until hits no longer became relevant to the research question. During this pilot it was felt that the first 50 hits captured most of the relevant websites. Furthermore we ran multiple different grey literature searches using different combinations of synonyms which allowed us to adequately capture a breath of resources. We also used snowball searching if any records were identified to be relevant. 

We have expanded on this in the methodology as follows: “We also used Google searches to identify any height/length measurement devices that were commercially available at the time of searching. These were run in incognito mode, with location and trending searches turned off. These searches used various combinations of structured search terms and synonyms similar to the formal literature search, and we inspected the first 50 hits. Targeted searches were also conducted on manufacturer websites looking for additional manufacturing information on devices in the academic literature. Snowball searching was also done if any records that were identified appeared relevant.”

We do however acknowledge that this approach comes with limitations and as such have included an additional section in our discussion to highlight this “multiple different combinations of synonyms in our grey literature search found a diversity of results, however we must also acknowledge that reviewing only the first 50 hits would mean that some devices could have been missed”. 

Would it be possible to indicate which reviewers conducted the reviews and which reviewer was approached for consultation?

Response: Yes, we have added in the initials of the reviewers as follows: 

“Two reviewers (TS and S Huang) independently assessed titles and abstracts of all retrieved citations. For potentially eligible studies, full texts were retrieved and assessed by two independent reviewers (TS and S Huang) according to the eligibility criteria. Disagreements during both stages were resolved either through discussion or consultation with a third reviewer (AW).”

Results:

Since the Abstract and Discussion note that 13 portable devices were identified, this should appear within the Results section. It can be determined by counting Table 2, but only that 16 studies were identified is mentioned in the Results. A sentence stating 13 portable devices would improve the manuscript clarity.

Response: Twelve portable devices were identified, one of which was trialled both in a controlled environment and in the field. An additional sentence stating all these devices has been added to the results, as follows:

 “Twelve height/length measurement devices were described across the 20 studies (Table 2). These devices included; SCANTIFY, Leica Disto, AutoAnthro, Gulliver, GLM40, Cannon digital camera, unspecified digital camera, P2B2, Optisizer, Coolpix, GLM30 and PePA” 

To assess the UNICEFF TPP, it seems that several characteristics (e.g., cost) go beyond those mentioned as part of in the customized data extraction form. How were these characteristics captured?

Response: Characteristics were captured by directly emailing manufacturers and utilising a grey literature search to look for specifications of devices found in the academic literature. This was described in the methods section as follows:

 “We also used Google searches to identify any height/length measurement devices that were commercially available at the time of searching.... Targeted searches were also conducted on manufacturer websites looking for additional manufacturing information on devices in the academic literature. …We directly emailed manufacturers where further clarification on technical specification was required”. 

Had the researchers come across the following study in their searches?

Additional study:

Baxter, J.-A., Roth, D., Zlotkin, S., Yin, S., Milgram, P. and Mouzaki, M. (2015), Measuring Child Length and Height: Assessing the Accuracy of a Portable Infrared-based Digital Tool. The FASEB Journal, 29: 31.3. https://doi.org/10.1096/fasebj.29.1_supplement.31.3

Response: Yes, this paper did come up in our initial search findings. It is a conference abstract with no corresponding full paper. At the time we contacted the authors to try and find the complete paper without success. The initial decision was made to exclude it on the basis of very limited information. However on reflection, we have amended the manuscript to include it as an abstract only paper. While it doesn’t add much depth to the paper due to very limited detail, adding it highlights a thorough search strategy, which allowed us to provide a solid and evidenced based review.

For completeness, the researchers should consider adding a simple table to the Appendix that lists the articles excluded and reasons for exclusion. These studies may be of interest to the readership.

Response: please see included as appendix 5

 Discussion:

Line 284-85: Per above, it is noted that 13 portable devices were identified. This had not previously been noted in the Results, only the number of studies. Suggest adding number of unique devices within the results, as 16 studies is mentioned, and determining that there are 13 devices requires counting lines in the table.

Response: please see comment above in results section. 

The Neale et al study is introduced in line 310, but then 2 sentences from line 311-16 (starting with UNICEF have identified...) appear before mention of Neale again. Are these 2 sentences supposed to appear here? They seem out of place. As well, being clear about how the methods/scope of the current review is different from Neale would improve the interpretability.

Response: Thank you we have outlined how our review differs to Neale et al and have revised for clarity as follows:

“Recent reviews looking at portable, digital measurement technology for neonates, infants and children have reported similar results. A systematic review by Van Gils et al looked at portable, digital technology for stress-free growth monitoring in preterm infants (38). Like our review, they found that novel digital technologies show high promise, yet current devices need ongoing refinements and improvements in order to meet gold standard requirements, especially within clinical contexts. Similarly, a narrative review by Neale et al assessed recent innovations in portable height/length measurement devices (39), concluding that manual measurement devices such as measurement mats and portable stadiometers remain the preferred measuring tools until digital equivalents can be shown to be accurate and precise. Our study had a wider search strategy scope allowing for numerous new devices to be found compared to the other reviews. However, similar to these reviews, we identified a large gap in the current digital product market, despite devices showing promise none currently meet the UNICEF TPP criteria for an ideal product.”

Based on the UNICEF TPP findings and last few paragraphs of the Results, could the researchers provide further critical reflection in the Discussion about the practicality of tools available, if they were to be used in diverse settings? (i.e., LMICs) A good portion of space in the Introduction reflects on stunting and wasting, and which are conditions more likely to affect those in LMICs. Do these devices seem practical for use in LMICs? For example, could reflect on cost relative to existing standard tools (could look at seca or Harpenden for cost comparison) or power requirements (standard length and height boards do not currently require a power source).

Response: Thank you. We touched on this issue in the discussion but have revised for clarity as follows:

“Furthermore, current cost of digital portable devices is significantly higher than a stadiometer. Additionally, operational requirements of these conventional methods such as stadiometers not requiring a power source can be considered beneficial, particularly in LMICs.”

Could the researchers reflect further on the limitations of their own study methods as conducted?

Response: Thank you we have now added further content to the limitations section of our discussion to the points outlined by the reviewer below:

(1) Given this is a review of the literature, the researchers likely didn't interact with any of the tools themselves. In other words, without ever seeing the tools, and just reading text, do the reviewers feel that they were adequately able to assess TPP characteristics? Comparing reports of some characteristics in the UNICEFF TPP could be subjective, especially given comparison is being made between studies.

 “During the TPP matching analysis, the reviewers did not test the discussed devices first hand, relying only on the literature for the analysis, limiting the assessment” 

(2) Measuring length and height is quite different, length is conventionally much more difficult to measure - largely related to the age of the child and their cooperation whilst being measured. There is information on length versus height in the Results, so this could be further reflected upon.

 “Given these devices were not directly tested, it was difficulty to extrapolate from the studies the likely added difficulties in measuring length of young children lying down and how this affected device usage.”

(3) As the UNICEF TPP was published in 2017 and several identified studies were published before this, it is quite plausible that they would not report on such characteristics. This could be reflected on further.

 “Additionally, it must be noted that many studies were written prior to the TPP publication in 2017, making it plausible that these studies may not have considered some specifications desired by the TPP, impacting the devices overall score.”

Appendix

The Tables presented in the Appendix do not align with the main manuscript (Table 1 in the Appendix is noted as Table 3 in text).

Response: Amended accordingly. Thank you.

Search strategy is provided in the Appendix, but not listed in the main paper. Within the Appendix, given the placement of mention of the search strategy, this should appear first (i.e., before UNICEF TPP)

Response: Amended accordingly. Thank you.

A table describing exclusions at full-text stage would strengthen the manuscript, and is standard practice for other review platforms

Response: Amended – added as appendix 5.

---

## [Decision Letter · Decision Letter 2]

10 Jul 2023

Portable digital devices for paediatric height and length measurement: a scoping review and target product profile matching analysis

PONE-D-22-30401R2

Dear Dr. Soller,

We’re pleased to inform you that your manuscript has been judged scientifically suitable for publication and will be formally accepted for publication once it meets all outstanding technical requirements.

Kind regards,

Nandita Perumal, PhD

Harvard TH Chan School of Public Health

Guest Editor

PLOS ONE

Reviewers' comments:

Reviewer's Responses to Questions

**Comments to the Author**

1. If the authors have adequately addressed your comments raised in a previous round of review and you feel that this manuscript is now acceptable for publication, you may indicate that here to bypass the “Comments to the Author” section, enter your conflict of interest statement in the “Confidential to Editor” section, and submit your "Accept" recommendation.

Reviewer #4: All comments have been addressed

2. Is the manuscript technically sound, and do the data support the conclusions?

Reviewer #4: Yes

3. Has the statistical analysis been performed appropriately and rigorously? 

Reviewer #4: Yes

4. Have the authors made all data underlying the findings in their manuscript fully available?

Reviewer #4: Yes

5. Is the manuscript presented in an intelligible fashion and written in standard English?

Reviewer #4: Yes

6. Review Comments to the Author

Reviewer #4: Thank you to the authors for addressing all items raised. This paper will be of interest to those in the field.

7. PLOS authors have the option to publish the peer review history of their article (what does this mean?). If published, this will include your full peer review and any attached files.

Reviewer #4: No

---

## [Editor Report · Acceptance letter]

17 Jul 2023

PONE-D-22-30401R2 

Portable digital devices for paediatric height and length measurement: a scoping review and target product profile matching analysis 

Dear Dr. Soller:

I'm pleased to inform you that your manuscript has been deemed suitable for publication in PLOS ONE. Congratulations! Your manuscript is now with our production department. 

Kind regards, 

on behalf of

Dr. Nandita Perumal 

Guest Editor

PLOS ONE